# Proton-coupled transport mechanism of the efflux pump NorA

Jianping Li[1], Yan Li[2], Akiko Koide [3,4], Huihui Kuang[5], Victor J. Torres [6,7], Shohei Koide [3,8], Da-Neng Wang [2] ✉ & Nathaniel J. Traaseth [1] ✉

Efflux pump antiporters confer drug resistance to bacteria by coupling proton import with the expulsion of antibiotics from the cytoplasm. Despite efforts there remains a lack of understanding as to how acid/base chemistry drives drug efflux. Here, we uncover the proton-coupling mechanism of the *Staphylococcus aureus* efflux pump NorA by elucidating structures in various protonation states of two essential acidic residues using cryo-EM. Protonation of Glu222 and Asp307 within the C-terminal domain stabilized the inward-occluded conformation by forming hydrogen bonds between the acidic residues and a single helix within the N-terminal domain responsible for occluding the substrate binding pocket. Remarkably, deprotonation of both Glu222 and Asp307 is needed to release interdomain tethering interactions, leading to opening of the pocket for antibiotic entry. Hence, the two acidic residues serve as a "belt and suspenders" protection mechanism to prevent simultaneous binding of protons and drug that enforce NorA coupling stoichiometry and confer antibiotic resistance.

Drug efflux is the broadest antibiotic resistance mechanism used by bacteria to survive in the presence of bacteriostatic and bactericidal compounds[1–3]. Expulsion of drugs is catalyzed by integral membrane proteins that harness ATP hydrolysis or secondary ion gradients[4]. The latter category of antiporters commonly uses aspartate and glutamate residues in membrane-embedded regions to bind protons and couple with the proton motive force (PMF)[5–11]. To ensure drug efflux is coupled to the energy gradient, idealized antiport must follow certain 'rules', including conformational exchange for substrate-bound states only and binding to protons or drug (but not both)[12–14]. Despite the progress in transporter structural biology and functional studies[15–21], the molecular basis of proton-coupled transport remains an open question in the field.

In this work, we studied *S. aureus* NorA as a model efflux pump antiporter from the major facilitator superfamily (MFS)[22–24]. NorA contains two essential membrane embedded acidic residues at positions Glu222 and Asp307 which are conserved across *S. aureus* strains and required for conferring fluoroquinolone resistance to the human pathogen methicillin-resistant *S. aureus* (MRSA)[25]. Drug efflux by NorA enables evolution of antibiotic target mutations to topoisomerases while inhibition of the transporter, likely through interactions with the acidic residues in the substrate binding pocket, blocks higher level mutations[26]. NorA predominantly populates the outward-open conformation at pH 7.5 as inferred from cryo-EM structures of NorA solved in complex with two different Fabs[25]. The proximity of Glu222 and Asp307 within NorA to an arginine residue in CDRH3 of the Fabs suggested these acidic residues were in their deprotonated forms. However, like other efflux systems relying on acidic residues for transport, the mechanism for how proton binding and release leads to drug efflux in NorA is unknown.

[1]Department of Chemistry, New York University, New York, NY, USA. [2]Department of Cell Biology, New York University School of Medicine, New York, NY, USA. [3]Perlmutter Cancer Center, New York University School of Medicine, New York, NY, USA. [4]Department of Medicine, New York University School of Medicine, New York, NY, USA. [5]Simons Electron Microscopy Center, New York Structural Biology Center, New York, NY, USA. [6]Department of Microbiology, New York University School of Medicine, New York, NY, USA. [7]Antimicrobial-Resistant Pathogens Program, New York University School of Medicine, New York, NY, USA. [8]Department of Biochemistry and Molecular Pharmacology, New York University School of Medicine, New York, NY, USA. ✉e-mail: da-neng.wang@med.nyu.edu; traaseth@nyu.edu

Here, we investigated the catalytic role of acid/base chemistry within NorA using cryo-electron microscopy, molecular dynamics simulations, binding assays, and MRSA growth inhibition experiments. Wild-type NorA and three mutants of the acidic residues were used to benchmark the protonation states of the transporter, thereby enabling structural studies to be correlated with the proton-coupling mechanism. Our findings reveal an elegant mechanism for how antiporters satisfy the transport rule of binding to only protons or drug.

## Results

### Inward-occluded structure of NorA

To determine whether acid/base chemistry at Glu222 and Asp307 modulated NorA's conformation, we performed binding experiments to Fab36, an antibody previously reported to stabilize the outward-open conformation of NorA[25]. Unlike results at pH 7.5, we discovered NorA did not bind to Fab36 at pH 5.0, indicating a unique conformation induced by the lower pH value (Supplementary Fig. 1a). We deduced residues Glu222 and Asp307 of NorA were predominantly protonated at pH 5.0 by showing that a double mutant mimicking the protonated state (NorA$^{E222Q,D307N}$) was unable to bind Fab36 at pH values of 5.0 or 7.5 (Supplementary Fig. 1a–c). Hence, to elucidate the structure of this conformation by cryo-EM, we identified a new Fab (FabDA1) using phage display against NorA$^{E222Q,D307N}$ to ensure binders to the protonated state. Both wild-type NorA and the double mutant bound FabDA1 with a low micromolar affinity at pH 5.0 but only NorA$^{E222Q,D307N}$ bound at pH 7.5 (Supplementary Fig. 1a–c), consistent with FabDA1 being a conformation selective binder.

NorA-FabDA1 and NorA$^{E222Q,D307N}$-FabDA1 were purified as 1:1 complexes at pH 5.0 and 7.5, respectively, and used for collection of large cryo-EM datasets. Subsequent processing yielded cryo-EM maps at 3.26 Å resolution for NorA and 3.61 Å resolution for NorA$^{E222Q,D307N}$, which contained the CDRH3 loop or the variable portion of FabDA1 following local refinement (Supplementary Figs. 2–4; Supplementary Table 1). Although some preferred orientation and flexibility were observed within the complexes, all the side chains could be unambiguously assigned using the high-quality cryo-EM maps (Supplementary Figs. 3 and 4).

Structures of NorA and NorA$^{E222Q,D307N}$ each displayed the inward-occluded conformation (backbone $C_\alpha$ r.m.s.d of 0.82 Å), where the substrate binding pockets were partially occluded from the cytoplasmic side of the membrane and fully closed from the periplasmic side (Fig. 1a, b). The N-terminal and C-terminal domains (NTD and CTD), comprised of TM1 to TM6 and TM7 to TM12, were rotated by ~50° relative to the outward-open conformation (Supplementary Fig. 5a). Underlying the transition between inward-occluded and outward-open conformations were changes at the conserved motifs A, B, and C[24]. Namely, the interaction between Asp63 in TM2 (part of motif A) and Arg324 in the linker between TM10 and TM11 was disrupted in the inward-occluded state relative to that in the outward-open state (Supplementary Fig. 5b). Furthermore, Arg98 in TM4 (part of motif B) displayed interactions within 4 Å to backbone carbonyls of Gly18 and Ile19 in TM1 for the inward-occluded state that were disrupted in the outward-open conformation. We also observed a more pronounced bending of TM5 at Gly143 and Pro144 of motif C for the inward-occluded state compared to the outward-open structure. Altogether, these observations indicated a "rocker switch" exchange mechanism[13,20,27–30] for substrate transport and supported the inward-occluded classification of NorA in complex with FabDA1. Lastly, it is notable that the CDRH3 loop of FabDA1 bound NorA at TM2, TM4, TM11, and the long loop between TM6 and TM7 (Supplementary Fig. 5c). Since this complex occurred on the cytoplasm surface of NorA and not within the substrate binding pocket, these interactions are unlikely to affect NorA's overall conformation.

For the inward-occluded conformation, TM5 of the NTD was primarily responsible for obstructing the entrance to the substrate binding pocket through interactions with TM7, TM8, TM10, and TM11 of the CTD (Fig. 1b, c). Carboxyl groups of Glu222 and Asp307 tethered TM7 and TM10 to TM5 through hydrogen bonds to backbone carbonyls of Phe140 and Ile141 and to the side chain of Asn137, respectively (Fig. 1c; Supplementary Fig. 5d). Surrounding these interactions inside the substrate binding pocket was a notable distribution of hydrophobic residues, including Ile23 from TM1, Ala134, Phe140, Ile141, and Pro144 from TM5, Leu218 from TM7, Ile244 and Ala252 from TM8, and Phe306 from TM11 (see yellow surface in Fig. 1c). While Glu222 was deeply embedded inside the substrate binding pocket and solvent inaccessible, Asp307 was more accessible from the cytoplasmic side of the membrane (Fig. 1d). Nevertheless, the hydrophobicity surrounding these residues likely promotes the occluded conformation by excluding bulk water and reduces the effective dielectric to strengthen the electrostatic interactions involving Glu222 and Asp307. Similar tethering interactions and hydrophobic environment were observed in the NorA$^{E222Q,D307N}$-FabDA1 complex, albeit with contacts mediated by the mutated residues, Gln222 and Asn307 (Supplementary Fig. 5d). The similarity of interactions with TM5 and overall structural correspondence between wild-type NorA and double mutant structures supported the conclusion that Glu222 and Asp307 were in the protonated forms in the wild-type NorA structure and that the E222Q and D307N mutations suitably mimicked the protonated forms.

### Protonation of Glu222 or Asp307 independently traps the occluded state

To determine whether protonation at Glu222 and/or Asp307 were responsible for triggering the conformational change to the inward-occluded state, we constructed the E222Q and D307N single mutants of NorA (NorA$^{E222Q}$ and NorA$^{D307N}$). Remarkably, binding assays revealed each mutant bound FabDA1 at pH 5.0 and 7.5 with $K_d$ values in the low micromolar range and neither bound to the outward-open-specific Fab36 at either pH value (Supplementary Fig. 1d, e). Next, cryo-EM samples of NorA$^{E222Q}$ and NorA$^{D307N}$ in complex with FabDA1 were purified at pH 7.5, which we anticipated would induce deprotonation at Asp307 or Glu222, respectively. Following data collection and processing, we obtained cryo-EM maps at 3.35 Å resolution for NorA$^{E222Q}$ and 3.51 Å resolution for NorA$^{D307N}$ (Supplementary Fig. 6). While some side chain density in the NorA$^{D307N}$ cryo-EM map was missing, it did not affect model building for residues in the substrate binding pocket (Supplementary Fig. 7).

Each single mutant structure displayed an inward-occluded conformation. Superimposition with wild-type NorA gave backbone $C_\alpha$ r.m.s.d. values of 0.73 Å for NorA$^{E222Q}$ and 1.07 Å for NorA$^{D307N}$, as well as similar interfacial contacts with CDRH3 of FabDA1 (Fig. 2a, b; Supplementary Fig. 5c). Nevertheless, small but significant structural changes were observed within the substrate binding pocket for NorA$^{D307N}$ and NorA$^{E222Q}$ structures relative to NorA, providing clues into the proton-coupling mechanism of drug efflux. The most significant differences were observed for NorA$^{D307N}$, which displayed movements of TM5 by 1–2 Å (Fig. 2a; Supplementary Fig. 5e). These changes were accompanied by 0.8 to 1.7 Å longer distances between Glu222 and the backbone carbonyl of Phe140 relative to distances in the NorA and NorA$^{E222Q,D307N}$ structures (Fig. 2c–e; Supplementary Fig. 5d). The distance change relative to NorA in the protonated form suggested Glu222 was deprotonated in the NorA$^{D307N}$ structure and was consistent with an ionized carboxyl group acting as a poorer hydrogen bond donor relative to its protonated state. We also observed a chi1 rotamer change of ~67° for Phe140 (Supplementary Fig. 5d, f), which may act to further destabilize the backbone hydrogen bond due to repulsion with the negatively charged Glu222 residue. On the other side of the tether, the nearest distance between Asn307 (mutant) and Asn137 was 1.5 to 2.1 Å longer relative to those in the NorA and NorA$^{E222Q,D307N}$ structures (Fig. 2c–e). Hence, a disrupted interaction at one site influenced the distance at the other, an observation consistent

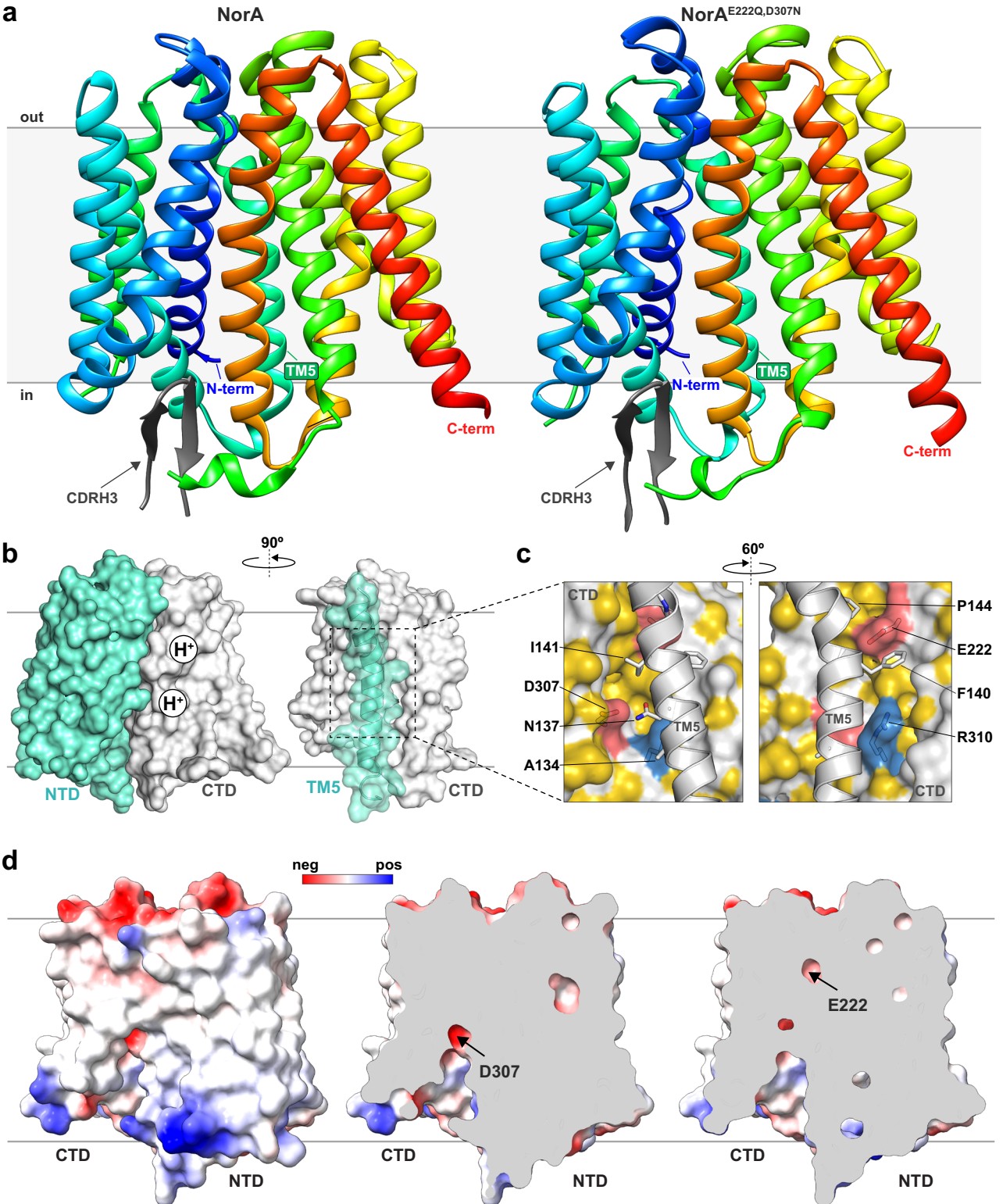

**Fig. 1 | Inward-occluded structures of NorA determined using cryo-EM.**
**a** Structures of NorA at pH 5.0 (left) and NorA$^{E222Q,D307N}$ at pH 7.5 (right) determined in complex with FabDA1. The structure at pH 5.0 corresponds to NorA protonated at Glu222 and Asp307. NorA TM helices are colored in rainbow, and CDRH3 of FabDA1 is displayed in grey. **b** Left: surface representations of NTD corresponding to TM1 to TM6 (in spring green) and CTD corresponding to TM7 to TM12 (in light grey). The two proton binding positions at Glu222 and Asp307 within the CTD are schematically depicted. Right: surface representations of TM5 (spring green) packing against the CTD (in light grey). **c** Expanded view of interactions between TM5 (in light grey) and Glu222 and Asp307 within the CTD as depicted in a partially transparent surface with select residues displayed in sticks. The surface representation is colored as follows: hydrophobic residues in yellow, anionic residues in red, cationic residues in blue, and all other residues in light grey[65]. **d** Surface and cutaway views of the electrostatic potential plotted on NorA. The sites of Glu222 and Asp307 are indicated in NorA. Blue, red, and white colors correspond to positively, negatively, and neutrally charged sites, respectively.

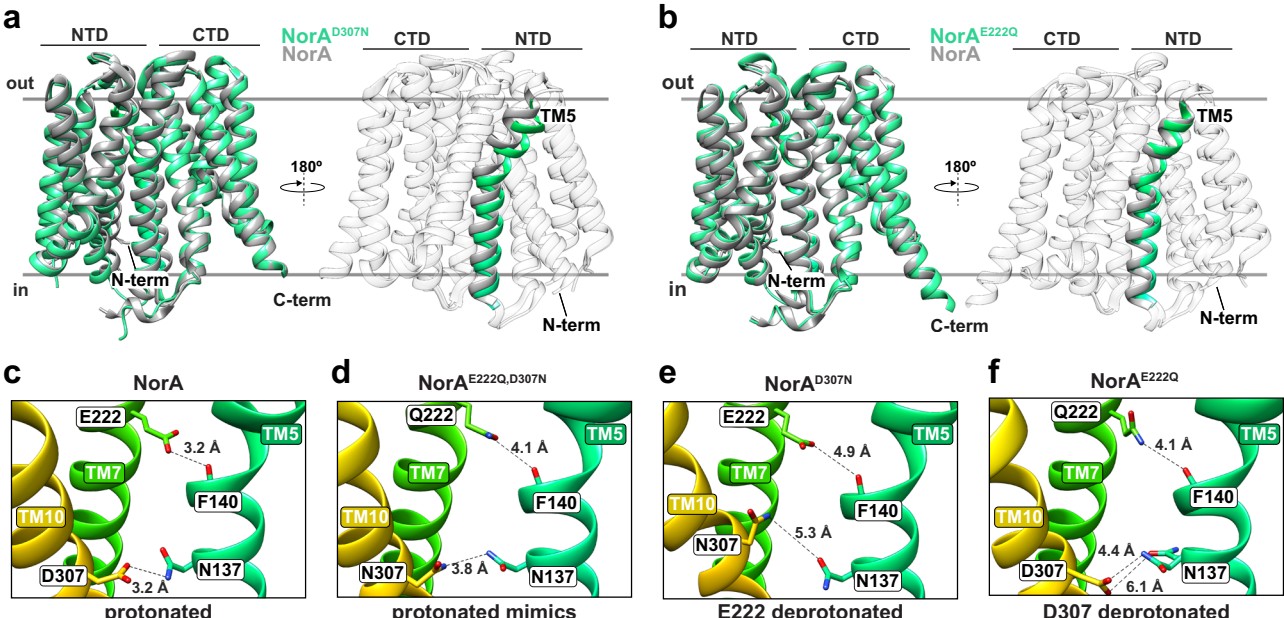

**Fig. 2 | Deprotonation of Glu222 or Asp307 displays subtle structural changes.**
**a**, **b** Left: cryo-EM structures of NorA^D307N (**a**) and NorA^E222Q (**b**) determined at pH 7.5 in complex with FabDA1 (in spring green) superimposed with the NorA structure determined at pH 5.0 (in grey). Right: same superimposition as in the left panel, except with TM5 highlighted (in spring green) and the structure rotated by 180°.
Other helices are colored in white and made partially transparent. **c**–**f** Nearest distances between Glu222 or Gln222 in TM7 (in green) to the backbone carbonyl oxygen of Phe140 in TM5 (in spring green) and Asp307 or Asn307 in TM10 (in gold) to Asn137 in TM5 for NorA (**c**), NorA^E222Q,D307N (**d**), NorA^D307N (**e**), and NorA^E222Q (**f**). Dashed black lines correspond to the indicated distances (in Å).

with the two-pronged tethering model for NorA protonated at Glu222 and Asp307.

A similar trend was identified in the NorA^E222Q structure, albeit to a lesser extent than in the NorA^D307N structure. The nearest distances of Gln222 (mutant) to the backbone carbonyl of Phe140 and Asp307 to the side chain of Asn137 were ~1 Å longer than in the NorA structure (Fig. 2c, f). Furthermore, two rotamers of Asn137 were observed in the NorA^E222Q cryo-EM map and were indicative of heterogeneity and a weaker interaction to Asp307 upon deprotonation (Fig. 2f; Supplementary Fig. 5d). Taken together with the NorA^E222Q structure, these findings indicated that single deprotonation of Glu222 or Asp307 was sufficient to stabilize the inward-occluded conformation while at the same time weakening the tethering interactions with TM5. This structural insight hinted that double deprotonation would have a more pronounced destabilizing effect on the inward-occluded conformation.

**Double deprotonation opens the substrate binding pocket**
To determine how deprotonation at both Glu222 and Asp307 alters the inward-occluded structure, molecular dynamics (MD) simulations in explicit lipid bilayers were carried out. Simulations were first benchmarked by investigating protonation of Glu222 and/or Asp307 starting from the wild-type NorA structure at pH 5.0. Simulations with both residues in the proton-bound states displayed no significant changes relative to the inward-occluded conformation, including preservation of tethering interactions involving Glu222 and Asp307 with TM5 observed in the cryo-EM structure (Supplementary Fig. 8a–d). This similarity supported the conclusion that FabDA1 binding on the cytoplasm side imparted no significant perturbation to the inward-occluded conformation of NorA. Simulations where NorA was singly protonated at Glu222 or Asp307 also maintained the inward-occluded conformation with no transitions to the inward-open conformation (Supplementary Fig. 8a–d). However, we observed increased distances between the carboxyl of Glu222 and the backbone carbonyl of Phe140 and the carboxyl of Asp307 and the amide of Asn137 in simulations involving deprotonation at Glu222 or Asp307 (Supplementary

Fig. 8b, c). These findings matched the cryo-EM structural findings on NorA^E222Q and NorA^D307N showing that deprotonation of Glu222 or Asp307 partially repelled TM5 from the substrate binding pocket.

Next, we initiated simulations for the apo form of NorA by deprotonating both Glu222 and Asp307 within the wild-type NorA inward-occluded structure. Although the total simulation time of 6 µsec did not capture the transition between the inward-occluded and outward-open conformations typically occurring on the msec to sec timescale[31], our results revealed a rather striking conformational change upon deprotonation of Glu222 and Asp307. In fact, opening of the substrate binding pocket occurred in two of the three replicate runs and coincided with the disruption of hydrogen bonds between Glu222 and the backbone carbonyl of Phe140 and between the side chains of Asp307 and Asn137 (Fig. 3a; Supplementary Fig. 8a–d). The loss of tethering interactions induced movement of TM5 by ~6 Å, resulting in water molecules entering the substrate binding pocket and residing in proximity to Glu222 and Asp307 (Fig. 3b, c; Supplementary Fig. 8e–g). This conformational change corresponded to the transition from the inward-occluded state to the inward-open state, the latter resembling a structure predicted using AlphaFold2 (Supplementary Fig. 8a). In addition to changes at Glu222 and Asp307, we observed residues Lys125, Phe129, Ser318 and Asn325 on the cytoplasmic face (termed the "cytoplasmic rim"[32]) to display altered interactions upon pocket opening that suggested a role in gating between inward-occluded and inward-open conformations (Supplementary Fig. 8h). Taken together with the cryo-EM structures and binding assays, we conclude that the two acidic residues use a "belt-and-suspenders" model where deprotonation of both Glu222 and Asp307 are required to disengage the tether of the inward-occluded conformation to achieve accessibility of the substrate binding pocket (Fig. 4).

**Proton-coupled transport model**
*Why is double deprotonation of Glu222 and Asp307 required to destabilize the inward occluded conformation and open the substrate binding pocket?* We posited that the answer was to ensure only protons or an antibiotic substrate can bind to NorA, thereby achieving electrogenic

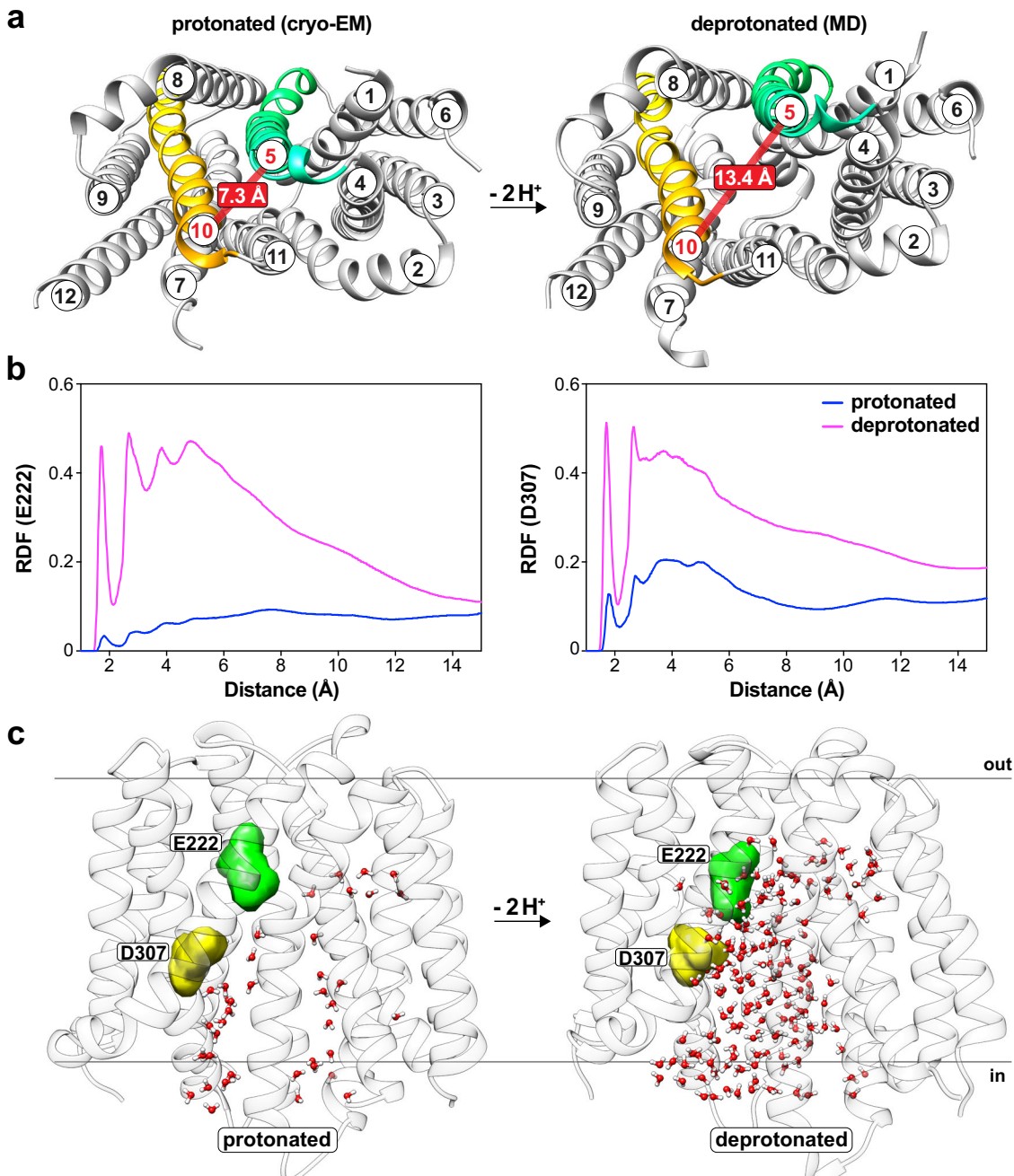

**Fig. 3 | Deprotonation of Glu222 and Asp307 induces the inward-open state in MD simulations. a** Cytoplasmic views of the NorA structure determined at pH 5.0 using cryo-EM (left) and NorA deprotonated at Glu222 and Asp307 derived from MD simulations (right). TM domain helices are numbered for reference, and TM5 (in spring green) and TM10 (in gold) are highlighted among other helices (in grey). The thick, red line corresponds to the indicated distance (in Å) between the Cα atoms of Ala126 in TM5 and Ser318 in TM10, illustrating the separation of TM5 from the CTD. **b** The water radial distribution function (RDF) surrounding Glu222 (left) or Asp307 (right) calculated from NorA MD simulations where Glu222 and Asp307 were protonated (blue) or deprotonated (magenta). **c** Representative snapshots from MD simulations of the water distribution in the substrate binding pocket of NorA protonated or deprotonated at Glu222 and Asp307. Water molecules are represented by red (oxygens) and white (hydrogens) spheres, Glu222 and Asp307 are highlighted by spring green and yellow surfaces, and TM helices are colored in light grey.

transport in a 2:1 proton:drug stoichiometry. Transport of a cationic drug with a +1 charge, a common substrate charge effluxed by NorA, would be electrogenic if two protons were transported and electroneutral if one proton was transported. In the former case, both the pH gradient ($\Delta pH$) and membrane potential can drive drug efflux, while the latter can only be driven by $\Delta pH$ (Fig. 5a). To corroborate structural findings suggesting two protons bound by Glu222 and Asp307 are transported, we performed growth inhibition experiments of MRSA

USA300 in the presence of ethidium bromide (+1 charge) by varying the external pH, which served to modulate $\Delta pH$ across the membrane. Results indicated that NorA conferred ethidium resistance over the entire pH range tested, displaying 7- to 15-fold higher $IC_{50}$ values relative to the control (Fig. 5b, c; Supplementary Fig. 9). Since $\Delta pH$ is zero or negative when the external pH is at or above 7.8[33–39], the resistance phenotype conferred by NorA under these conditions indicated the membrane potential alone was sufficient to drive

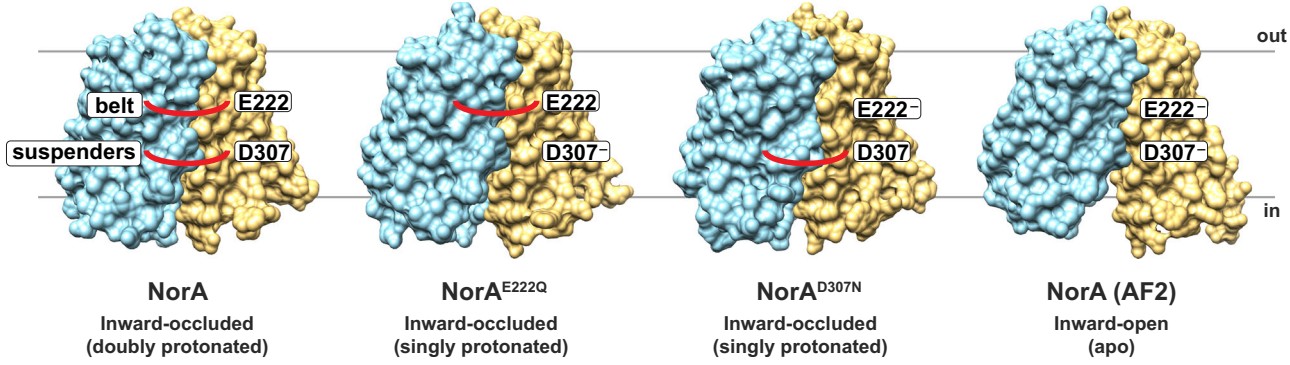

**Fig. 4 | Belt and suspenders model for NorA.** A schematic depiction of the "belt and suspenders" mechanism. The models show surface representations of inward-occluded cryo-EM structures of wild-type NorA at pH 5.0, NorA$^{E222Q}$ at pH 7.5, and NorA$^{D307N}$ at pH 7.5 and the inward-open model predicted from AlphaFold2 (likely to be apo based on MD simulations). Red lines connecting the NTD and CTD denote the "belt" or "suspenders". "E222$^-$" and "D307$^-$" correspond to deprotonated Glu222 and Asp307 residues and "E222" and "D307" correspond to protonated Glu222 and Asp307 residues. The NTD and CTD are colored in blue and orange, respectively.

transport. Therefore, these findings support the conclusion that NorA performs efflux in a 2:1 proton:drug stoichiometry (i.e., electrogenic transport) and harmonizes with the location of Glu222 and Asp307 as the only essential acidic residues embedded in the substrate binding pocket. Based on our structural and functional insights, we propose a transport cycle of NorA displaying the 2:1 proton:drug stoichiometry (Fig. 5d). This model underscores how Glu222 and Asp307 permit antibiotic binding by coupling proton release with opening of the substrate binding pocket.

### MRSA growth inhibition and efflux assays support the tethering model

According to our model (Fig. 5d), blocking the transition from the inward-occluded state to the inward-open state would disrupt antibiotic binding to NorA, thereby halting the transport cycle and ablating NorA function. Since our cryo-EM and MD simulation results implicated deprotonation of Glu222 and Asp307 in mediating the conformational change, we designed single-site mutants adjacent to these residues to test this hypothesis (Fig. 6a, b). Genes corresponding to wild-type *norA* and mutants were constructed on a hemin-inducible plasmid, transformed into a strain of MRSA where the native *norA* gene was disrupted by a transposon insertion (MRSA$^{\Delta norA}$)[40], and verified for protein expression (Fig. 6c). Growth inhibition assays in MRSA$^{\Delta norA}$ were performed by measuring the minimum inhibitory concentration (MIC) to norfloxacin, a fluoroquinolone substrate transported by NorA. MIC values for the point mutants NorA$^{E222Q}$ and NorA$^{D307N}$ displayed substantial loss-of-function, nearly identical to the MIC value of MRSA$^{\Delta norA}$, which underscored their role in proton-coupled transport (Fig. 6b). To complement these MIC measurements, we performed ethidium efflux assays on NorA, NorA$^{E222Q}$, and NorA$^{D307N}$ in MRSA$^{\Delta norA}$. The mutants displayed loss-of-function efflux activities close to that of MRSA$^{\Delta norA}$ and significantly reduced relative to wild-type NorA (Fig. 6d, e). These results were consistent with MIC observations and cryo-EM structures, showing that the mutations trap NorA in the inward-occluded conformation.

Other NorA mutants also exhibited significantly lowered MIC values in agreement with their interactions and proximity to Glu222 and Asp307. For example, the N137A mutant nearly ablated NorA's resistance phenotype, which was anticipated given the electrostatic interactions between Asn137 and protonated Asp307 in the cryo-EM structures. Namely, mutation of Asn137 to alanine would disrupt hydrogen bonds, thereby destabilizing the inward-occluded conformation and altering the transport cycle. Likewise, we observed a notable difference in MIC between I141A and I141Q mutants, which revealed that a smaller hydrophobic residue was somewhat tolerated

at this position, albeit with a ~2-fold reduced MIC relative to wild-type, while a similar sized residue with altered polarity nearly ablated NorA's activity. In the presence of the native Ile141 residue or the smaller alanine mutation, ionization of Glu222 is expected to repel from this hydrophobic residue to favor a conformational change toward the inward-open state. Such an opening of NorA's structure coincided with disruption of Glu222 interactions with TM5 and greater water penetration in the substrate binding pocket, as observed in MD simulations (Fig. 3; Supplementary Fig. 8e). However, the presence of the glutamine mutation at position 141 is expected to reduce the repulsive character and likely stabilize the inward-occluded conformation by forming hydrogen bonds with the ionized Glu222 residue. Such a stabilizing interaction would trap NorA in the inward-occluded conformation and reduce catalytic turnover leading to the loss-of-function phenotype for I141Q. Overall, these functional results on NorA mutants in the human pathogen MRSA support the tethering model and the importance of hydrophobic residues nearby Glu222 and Asp307 within the substrate binding pocket.

## Discussion

We uncovered the proton-coupling mechanism for NorA, including how deprotonation of two membrane embedded acidic residues triggers the opening of NorA's substrate binding pocket and yields a 2:1 proton:drug stoichiometry (Fig. 5d). Glu222 and Asp307 play analogous roles to a "belt and suspenders", each independently able to stabilize the inward-occluded conformation and achieve strict correspondence between binding of protons or drug. These findings offer a structural explanation in NorA for the vectorial transport rule that antiporters cannot bind protons and substrate concurrently[14].

The molecular forces regulating pocket opening likely stem from repulsion created upon ionization of acidic residues within the low dielectric environment of the substrate binding pocket. Indeed, the inward-occluded conformation is stabilized by hydrogen bonds between Glu222/Asp307 and TM5 and is surrounded by nearby hydrophobic residues, including Phe140 and Ile141. It appears nature uses acid/base chemistry as a simple mechanism to repel or attract interdomain interactions between the NTD and CTD in NorA (Fig. 4). Loss of activity observed for the I141Q mutation of NorA supports this theory and suggests the absence of hydrophobic amino acids surrounding the carboxyl groups may trap specific conformations in the catalytic cycle leading to reduced turnover. Since NorA achieves broad specificity to drugs varying in chemical structure, our findings suggest the hydrophobic character of its substrate binding pocket is dually optimized for broad specificity and repulsive forces following ionization of acidic residues within membrane embedded regions. The large

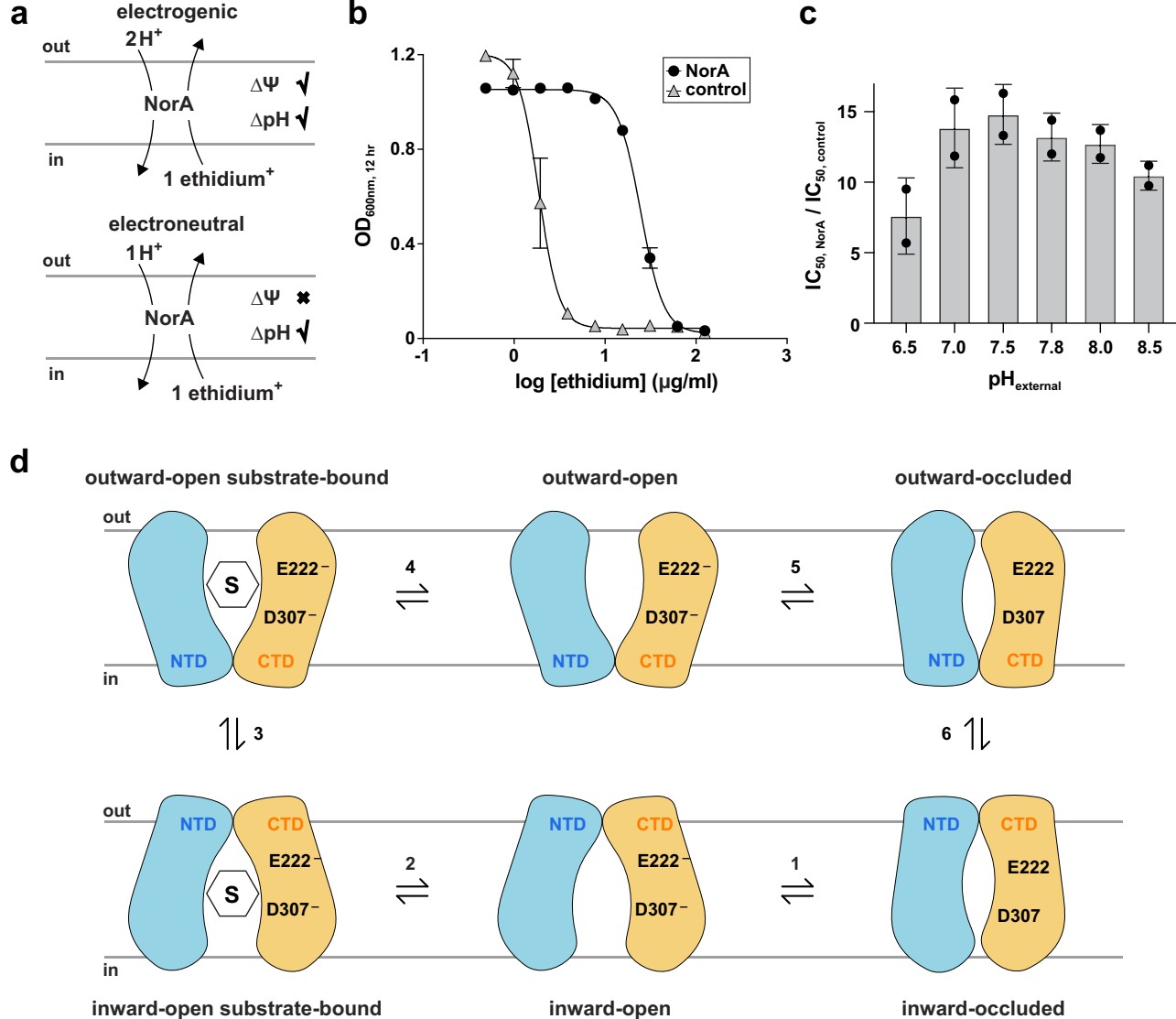

**Fig. 5 | NorA transports drug substrates with 2:1 H⁺:drug stoichiometry.**
**a** Diagram of NorA transport stoichiometries of 2:1 H⁺:ethidium (electrogenic, top) or 1:1 H⁺:ethidium (electroneutral, bottom) and the two energy sources for efflux: proton gradient (ΔpH) and membrane potential (Δψ). Electroneutral transport is driven by ΔpH, while electrogenic transport can be driven by ΔpH or Δψ. **b** Growth inhibition experiments at pH 7.8 of MRSA$^{\Delta norA}$ transformed with a hemin-inducible plasmid containing the wild-type *norA* gene (black circles) or a dead *norA* mutant control (grey triangles) in the presence of variable ethidium bromide concentration (0 to 125 μg/mL). Bacterial growth is displayed by the OD$_{600nm}$ value at the 12 hr timepoint. Solid lines correspond to non-linear fits of experimental data to determine the IC$_{50}$ values. The standard deviations are derived from four replicates obtained from two independent experiments. **c** IC$_{50}$ values of plasmids containing wild-type *norA* divided by the respective control plasmid as a function of different external pH values ranging from 6.5 to 8.5. Mean values ± s.d. are depicted for the IC$_{50}$ values. **d** Proposed NorA transport cycle. The NTD and CTD are colored in blue and orange, "S" is the drug substrate, "E222⁻" and "D307⁻" represents deprotonated Glu222 and Asp307, and "E222" and "D307" are protonated Glu222 and Asp307. Step 1: Deprotonation of Glu222 and Asp307 in the inward-occluded state releases the TM5 tether, leading to opening of the substrate binding pocket. Step 2: drug binds to the inward-open conformation of NorA from the cytoplasmic side of the membrane. Step 3: conformational exchange flips the drug-bound inward-open state to the outward-open state. Step 4: drug is released into the periplasm, resulting in the apo, outward-open conformation. Step 5: Glu222 and Asp307 bind protons in the periplasm to give an outward-occluded conformation. Step 6: NorA undergoes conformational exchange to the inward-occluded conformation, preparing it for the next round of substrate turnover.

number of antiporters carrying out electrogenic transport suggests our model may have relevance for explaining how other efflux systems achieve proton-coupled transport and confer drug resistance to pathogenic bacteria.

## Methods

### Expression and purification of NorA

The NorA expression and purification procedure was performed as previously described[25]. In brief, NorA was expressed in C43 (DE3) *E. coli* using the autoinduction method with slight modifications. Cells were grown at 32 °C in ZYP-5052 medium supplemented with 1 mM MgSO4. At an OD$_{600nm}$ of 0.5, the temperature was lowered to 20 °C, and cultures were grown for an additional 18–20 hours. Cells were collected, resuspended, and lysed in 40 mM Tris pH 8.0, 400 mM NaCl, and 10% glycerol. The membrane fraction was obtained by ultracentrifugation and solubilized with 1% (wt/vol) lauryl maltose neopentyl glycol (LMNG).

NorA was purified using immobilized metal-affinity chromatography, wherein it was bound to cobalt affinity resin (ThermoFisher), washed with buffer (20 mM Tris pH 8.0, 200 mM NaCl, 10% glycerol

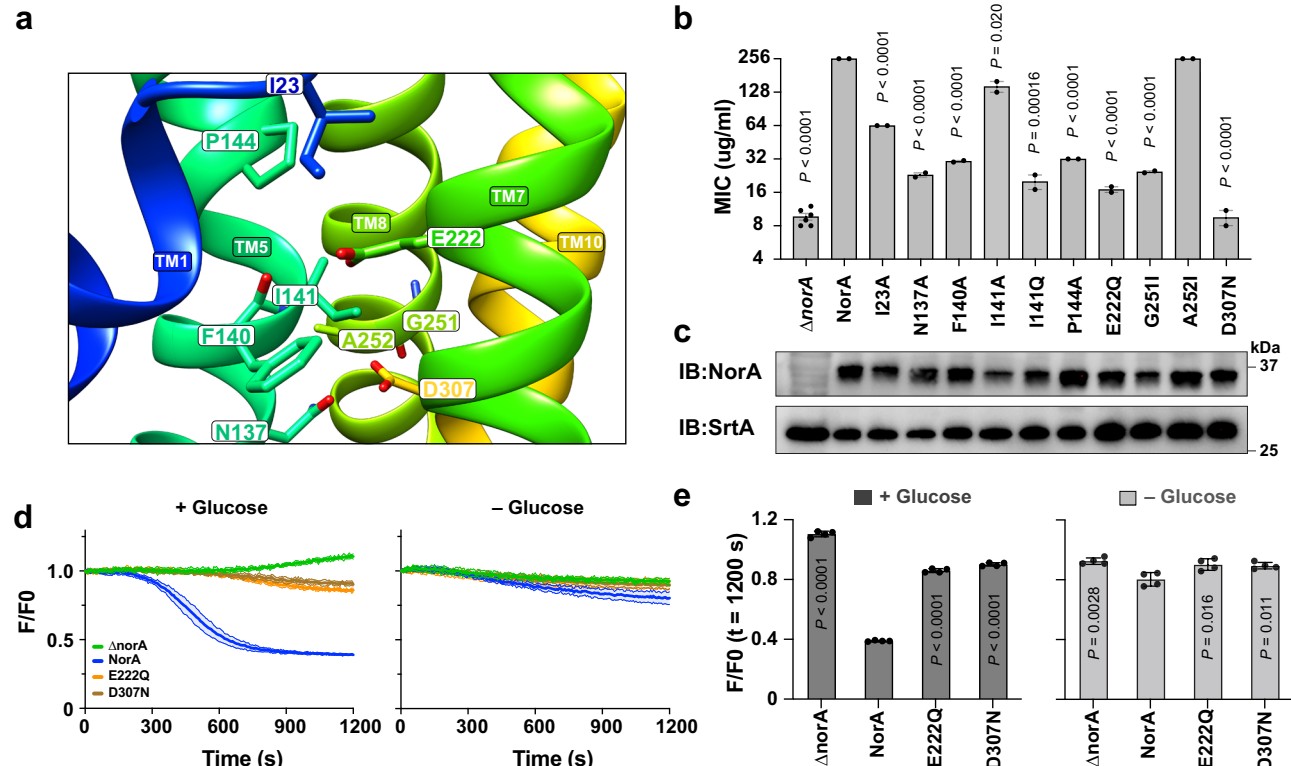

**Fig. 6 | Growth inhibition experiments in MRSA validate mechanistic findings.**
**a** View of the inward-occluded NorA cryo-EM structure at pH 5.0 displaying Glu222 and Asp307 and residues within 5 Å from these two residues. **b** Norfloxacin minimum inhibitory concentration (MIC) values were measured in an MRSA strain where the native *norA* gene was disrupted by a transposon insertion (MRSA$^{\Delta norA}$). Wild-type NorA and indicated mutants were encoded on a hemin-inducible plasmid. The number of independent MIC experiments (five for empty vector ∆*norA*, two for other variants) is represented by the black circles overlaid on the bar graph for each sample. Data are presented as mean values ± s.d. across independent experiments. The upper limit of MIC detection was set at 256 µg/mL. Statistical significance was assessed using an unpaired, two-tailed t-test for each single-site mutant in comparison to MRSA$^{\Delta norA}$ + NorA. *P* values were labeled on each bar. **c** Immunoblotting results performed on MRSA cell lysates following the induction of MRSA$^{\Delta norA}$ or MRSA$^{\Delta norA}$ transformed with a hemin-inducible plasmid carrying

wild-type NorA and NorA mutants. The immunoblots probed for the Myc tag on the C-terminus of NorA and SrtA (control protein in *S. aureus*). Similar results were observed in two independent experiments. **d** Ethidium efflux assay results for MRSA$^{\Delta norA}$ transformed with plasmids corresponding to empty vector (∆*norA*), wild-type NorA, E222Q, and D307N. The graph displays the normalized ethidium fluorescence (F/F0) over time, starting from the addition of glucose at time zero. Solid and shaded lines in the same color depict the mean values ± s.d. from two independent experiments with a total of four technical replicates. **e** A plot of the normalized fluorescence value (F/F0) at 1200 s from the data in **d**. The number of technical replicates from two independent experiments is represented by the black circles overlaid on the bar graph for each sample. Data are presented as mean values ± s.d. from four technical replicates. Statistical significance was evaluated using an unpaired, two-tailed t-test for individual single-site mutants compared to MRSA$^{\Delta norA}$ + NorA. *P* values were labeled on each bar.

and 0.2% (w/v) LMNG) containing varied imidazole concentrations, and eluted with 400 mM imidazole. Elution fractions were dialyzed into SEC buffer (20 mM Tris pH 7.5 and 100 mM NaCl) and incubated overnight with PMAL-C8 amphipol. Next, the sample was incubated with Bio-beads (100-fold excess by mass relative to total LMNG) and purified using a Superdex 200 10/300 column (Cytiva). The resulting peak fractions were pooled, concentrated, and utilized immediately or stored at −80 °C. For Fab screening, LMNG-purified NorA$^{E222Q,D307N}$ was reconstituted into lipid nanodiscs using biotinylated MSP1E3D1 following an established procedure[41], and the efficiency of reconstitution was assessed through SDS-PAGE.

**Identification of synthetic antibodies**
Sorting of a synthetic antibody phage-display library was performed essentially following a published procedure[25], with target concentrations of 100 nM (the first and second rounds), 50 nM (the third round), and 20 nM (the fourth round). In each round, the phage pool was incubated with streptavidin-coated magnetic beads saturated with a biotinylated MSP1E3D1 harboring wild-type NorA, and the unbound fraction was used for the sorting with biotinylated MSP1E3D1 harboring NorA$^{E222Q,D307N}$. The clone FabDA1 was identified from the sorted pool from the fourth round using phage MBBA method[42]. The FabDA1

gene with the rigidified elbow mutations[43] was cloned in an expression vector as described previously[25].

**Expression and purification of Fabs**
Fabs were expressed in TBG medium for 22 hours at 30 °C. Cell pellets were lysed using a high-pressure homogenizer in running buffer (20 mM sodium phosphate pH 7.0) containing 1 mg/mL hen egg lysozyme (Sigma). The supernatant, after centrifugation, was loaded onto a protein G column (Cytiva) pre-equilibrated with running buffer. Fab was eluted using 100 mM glycine (pH 2.7) and immediately neutralized using 2 M Tris buffer (pH 8.0). The eluted fractions with Fab were pooled, flash-frozen, and stored at −80 °C until further use.

**Microscale thermophoresis binding assays**
Purified Fabs were labeled with a primary amine-reactive 647-nm fluorophore (Nanotemper), and labeling efficiency was assessed by measuring absorbance at 280 and 650 nm. NorA embedded in amphipol underwent two rounds of SEC to remove free amphipol. Microscale thermophoresis (MST) was performed using the Monolith NT.115 Pico instrument (Nanotemper Technologies). NorA at varied concentrations was mixed with labeled Fab at a final concentration of 40 nM. Binding affinity measurements were conducted in buffers of

20 mM Tris pH 7.5, 100 mM NaCl, and 0.025% (wt/vol) Tween-20, or 50 mM acetate pH 5.0, 100 mM NaCl, and 0.025% (wt/vol) Tween-20, respectively. After incubation for 15 min, all samples were loaded into premium glass capillaries. $K_d$ values were obtained by plotting normalized fluorescent signal ($F_{norm}$) against ligand concentration and fitting with a non-linear sigmoidal regression model (GraphPad Prism) using a Hill slope equal to 1.0. All MST binding experiments were performed with three replicates.

### Preparation of cryo-EM samples

Purified NorA in amphipol was mixed with FabDA1 at a molar ratio of 1:3 and incubated for 1 hour at 4 °C. For the NorA-FabDA1 cryo-EM sample, a buffer of 50 mM acetate pH 5.0 and 100 mM NaCl was used, while a buffer of 20 mM Tris pH 7.5 and 100 mM NaCl was used for the NorA mutants in complex with FabDA1. SEC was performed to remove the free FabDA1 using a Superdex 200 10/300 column, and fresh fractions of the complex were immediately used to freeze the cryo-EM grids. The complex formation was verified using SDS-PAGE gel. For cryo-EM grid freezing, 4 μL of the sample at ~25 μM was applied to a glow-discharged UltrAuFoil 300-mesh R1.2/1.3 grid (Quantifoil). The grid was then blotted for 3.5 or 4 sec under 100% humidity at 4 °C before plunge-freezing into liquid ethane using a Mark IV Vitrobot (ThermoFisher Scientific).

### Cryo-EM data collection

Cryo-EM data collection for NorA-FabDA1 samples was performed using a Titan Krios microscope (ThermoFisher Scientific) equipped with a K3 direct electron detector (Gatan) and a GIF-Quantum energy filter with a 20 eV slit width. Automated data collection was carried out using Leginon 3.6[44]. Movies were recorded at a nominal magnification of 105,000× in SuperRes mode with a physical pixel size of 0.422/0.4124 Å and dose fractionated over 40 frames. The accumulated doses for the NorA-FabDA1 and NorA$^{E222Q,D307N}$-FabDA1 datasets were 50.25 e$^-$/Å$^{-2}$, while NorA$^{E222Q}$-FabDA1 and NorA$^{D307N}$-FabDA1 datasets were 44.13 e$^-$/Å$^{-2}$. In total, 12,953 movies were collected for NorA-FabDA1, 11,811 movies for NorA$^{E222Q,D307N}$-FabDA1, 8,884 movies for NorA$^{E222Q}$-FabDA1, and 10,461 movies for NorA$^{D307N}$-FabDA1.

### Cryo-EM data processing

The cryo-EM datasets were processed in cryoSPARC[45]. Imported movies were motion-corrected, and the contrast transfer function was estimated. Micrographs with an overall resolution worse than 5 Å were excluded from further analysis. Initial two-dimensional (2D) class averages were generated using particles selected from blob picking. Particles in 2D classes displaying features of NorA-Fab complexes were used as templates for Topaz picking[46,47]. A total of 8.18 million particles were Topaz selected from the NorA-FabDA1 dataset, 9.06 million particles from the NorA$^{E222Q,D307N}$-FabDA1 dataset, 6.80 million particles from the NorA$^{E222Q}$-FabDA1 dataset, and 6.99 million particles from the NorA$^{D307N}$-FabDA1 dataset. Particles underlying well-resolved 2D classes were used for initial ab initio model building, while all picked particles were used for subsequent heterogeneous three-dimensional (3D) refinement.

Particles corresponding to NorA-Fab complexes or NorA-Fab complexes lacking the Fab constant domain were kept for additional rounds of ab initio model building and heterogeneous refinement. After ~30 rounds of heterogeneous refinement, a round of non-uniform 3D refinement was applied to generate maps with overall resolutions of 3.66 Å for NorA-FabDA1 from 279,282 particles, 3.67 Å for NorA$^{E222Q,D307N}$-FabDA1 from 406,620 particles, 3.52 Å for NorA$^{E222Q}$-FabDA1 from 246,742 particles and 3.85 Å for NorA$^{D307N}$-FabDA1 from 136,284 particles, as assessed using the gold standard Fourier shell correlation (FSC). All maps were further processed using local refinement with a mask for NorA. The final local-refined maps were obtained with resolutions of 3.26 Å for NorA, 3.61 Å for NorA$^{E222Q,D307N}$, 3.35 Å for

NorA$^{E222Q}$, and 3.54 Å for NorA$^{D307N}$ in complex with FabDA1. Note that maps included FabDA1 density only for the CDRH3 loop in the NorA and NorA$^{E222Q}$ reconstructions and Fv for NorA$^{E222Q,D307N}$ and NorA$^{D307N}$ reconstructions. Sharpened maps were used for building structural models.

### Construction of structural models

The complex structural models were constructed using Namdinator[48] and Coot[49], and subsequently refined in Phenix[50]. The AlphaFold2 predicted inward-open structure model served as the reference to initiate NorA model building[51,52], with the flexible long loop region removed. The CDRH3 and F$_v$ parts were modeled starting from PDB ID 7LO7 and 7LO8[25].

### MIC measurement in MRSA

MRSA cells were cultured overnight at 37 °C in tryptic soy broth (TSB) medium containing 10 μg/mL chloramphenicol and 1 μM hemin. Next, the cells were diluted to ~1×10$^8$ colony-forming units (CFU) per mL and spread onto tryptic soy agar (TSA) plates with 5 μg/mL chloramphenicol and 0.4 μM hemin. Norfloxacin MIC test strips (Liofilchem) were placed in the center of the plate and incubated for 24 hrs at 37 °C. To prevent hemin degradation, all steps were performed in a dark environment. Two to five independent MIC experiments were carried out for each sample, and reported errors represent the standard deviation (s.d.) among independent experiments.

### Ethidium bromide efflux assay

MRSA cells were initially cultured overnight at 37 °C in tryptic TSB medium supplemented with 10 μg/mL chloramphenicol and 1 μM hemin. The cells were transferred to fresh TSB medium containing 5 μg/mL chloramphenicol and 1 μM hemin. Upon reaching mid-log phase (OD$_{600nm}$ of ~1.0) at 37 °C, the cells were subjected to four wash cycles using PBS buffer (50 mM phosphate pH 7.4, 137 mM NaCl, 2.7 mM KCl). The cells were resuspended in PBS buffer containing 5 μM carbonyl cyanide m-chlorophenyl hydrazine (CCCP) and 10 μg/mL ethidium bromide at an OD$_{600nm}$ of 0.6. Following resuspension, cells were incubated for 30 min at 37 °C with gentle shaking at 60 r.p.m. Next, cells were centrifuged for 10 min, and the resulting pellet was stored on ice until the fluorescence experiment. For the efflux assay, cells were resuspended in PBS with 10 mg/mL ethidium bromide, and the assay was initiated by addition of 0.2% (w/v) glucose. Control experiments were conducted under identical conditions, except without the addition of glucose. Fluorescence decay measurements were performed using a Molecular Devices FlexStation 3 instrument, with excitation at 530 nm and emission at 600 nm, over a period of 1200 sec. Each experiment was independently repeated twice, and error bars represent the standard deviation of duplicate experiments conducted on the same day.

### Characterization of NorA and mutant expression levels in MRSA

The MRSA$^{ΔnorA}$ strain was transformed with an empty vector or a hemin-inducible vector harboring NorA variants and cultured overnight at 37 °C in TSB medium containing 5 μg/mL chloramphenicol and 5 μM hemin. Subsequently, cells were normalized based on their optical density at 600 nm (OD$_{600nm}$) and lysed in 4× SDS sample buffer, which consisted of 8% SDS, 250 mM Tris at pH 6.8, 50% glycerol, 10% β-mercaptoethanol, and 0.3 mM bromophenol blue. Samples were loaded onto a 10% SDS-PAGE gel and subjected to immunoblotting. Primary antibodies specific to the C-terminal Myc tag on NorA (Genscript) and SrtA were used, with dilutions of 1:1000 and 1:20,000 for NorA and SrtA[53], respectively.

### NorA drug resistance assay under different pH conditions

MRSA$^{ΔnorA}$ strains were transformed with the pOS1-PhrtAB plasmid[54] containing *norA* or *norA$^{D307N}$* genes and plated on TSB agar. A single

colony was selected and grown in TSA medium supplemented with 10 μg/mL chloramphenicol and 1 μM hemin for ~19 hours at 37 °C. Subsequently, cultures were diluted by 200-fold into fresh TSB medium at various pH values (6.5, 7.0, 7.5, 7.8, 8.0, 8.5) and supplemented with 5 μg/mL chloramphenicol, 1 μM hemin, and varying concentrations of ethidium bromide (0 to 125 μg/mL). The optical density at 600 nm ($OD_{600nm}$) was measured every 15 min using a Bioscreen Pro C instrument at 37 °C with slow shaking.

## Molecular dynamics simulation

The MD simulations and data analysis were conducted using Gromacs 2020.4. Simulations were performed on NorA in various protonation states, namely protonated at Glu222 and Asp307, protonated at either Glu222 or Asp307, and deprotonated at Glu222 and Asp307. Each production simulation was conducted for 2 μsec and three replicas with three random velocities were executed for each protonation state for a total simulation time of 6 μsec. Coordinates were saved every 50 psec for analysis.

The setup of the simulations is described as follows. Starting from the inward-occluded cryo-EM structure of NorA determined at pH 5.0, the protein was embedded within a DOPG/DOPE lipid bilayer using CHARMM-GUI[55,56]. The missing loop between TM6 and TM7 in the structure was modeled using Modeller[57]. Subsequently, the system was solvated with TIP3P water molecules. To maintain electroneutrality, $Na^+$ ions were introduced: 44 $Na^+$ ions were added for protonated NorA, 45 $Na^+$ ions for singly protonated NorA, and 46 $Na^+$ ions for deprotonated NorA; details of the setup are displayed in Supplementary Table 2. The CHARMM36 force field was used for NorA[58], lipids[59], and TIP3P water[60].

The LINCs algorithm with a timestep of 2 fs was used to restrain hydrogen bonds to heavy atoms[61]. The particle-mesh Ewald method with a cutoff of 1.2 nm was employed to calculate electrostatic interactions[62], while van der Waals interactions were turned off within the range of 1.0 nm to 1.2 nm. A steepest descent minimization was performed on the system, ensuring that the maximum force on heavy atoms remained below 1000 kJ mol$^{-1}$ nm$^{-2}$ while keeping the heavy atoms constrained. After a 1 ns equilibration under constant volume and temperature, with heavy atoms constrained, a constant pressure and temperature simulation was conducted, gradually reducing the maximum force on heavy atoms to 0 kJ mol$^{-1}$ nm$^{-2}$ over 3 ns. Subsequently, 30 ns of MD simulations without constraints were carried out for equilibration (Supplementary Fig. 10) prior to commencing the production runs. Throughout the simulations, the system temperature was set at 298 K using the Nose-Hoover thermostat[63], which was independently applied to both the protein and solvent. The Parrinello–Rahman coupling method was utilized to maintain the pressure at 1 bar[64].

## Reporting summary

Further information on research design is available in the Nature Portfolio Reporting Summary linked to this article.

## Data availability

The data that support this study are available from the corresponding authors upon request. The cryo-EM maps have been deposited in the Electron Microscopy Data Bank (EMDB) under accession codes EMD-41605 for the EMDB entry of NorA; EMD-41606 for the EMDB entry of NorA$^{E222Q,D307N}$; EMD-41608 for the EMDB entry of NorA$^{D307N}$; EMD-41607 for the EMDB entry of NorA$^{E222Q}$. The atomic coordinates have been deposited in the Protein Data Bank (PDB) under accession codes 8TTE for the PDB entry of NorA; 8TTF for the PDB entry of NorA$^{E222Q,D307N}$; 8TTH for the PDB entry of NorA$^{D307N}$; 8TTG for the PDB entry of NorA$^{E222Q}$. The published NorA PDB codes 7LO7 and 7LO8[25] were referred to in this work. The input for MD simulations and the coordinates of starting and ending structures are accessible via

Zenodo [https://doi.org/10.5281/zenodo.11075313]. Supplementary Information is available for this paper. Source data are provided with this paper, corresponding to Figs. 3b, 5b, c, 6b–e, and Supplementary Figs. 1, 8b–f, 9.

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

## Acknowledgements

The authors are thankful for the following financial support: N.J.T. was supported by NIH (R01AI165782, R01AI108889) and NSF (MCB 1506420), D.N.W. was supported by NIH (R01NS108151, R01DK135088, R01GM121994, R01AI165782), and S.K. and V.J.T. were supported by NIH (R01AI165782). We thank J.F. Hunt for the MSP1E3D1-T277C construct, D.C. Hooper for sharing a NorA construct in the *pTrcHis2C* vector, A. S. John for assistance in the MIC measurements, and A. Besch and D. N. Brawley for helpful discussions. The following reagent was provided by the Network on Antimicrobial Resistance in *Staphylococcus aureus* (NARSA) for distribution by BEI Resources, NIAID, NIH: *Staphylococcus*

*aureus* subsp. aureus, Strain JE2, Transposon Mutant NE1034 (SAUSA300_0680), NR-47577. Screening of Cryo-EM grids was conducted at NYU Langone Health's Cryo-Electron Microscopy Laboratory (RRID: SCR_019202), which is partially supported by the Laura and Isaac Perlmutter Cancer Center Support Grant NIH/NCI P30CA016087. The large cryo-EM data collections were performed at the National Center for Cryo-EM Access and Training (NCCAT) and the Simons Electron Microscopy Center located at the New York Structural Biology Center, supported by the NIH Common Fund Transformative High-Resolution Cryo-Electron Microscopy program (U24 GM129539), and by grants from the Simons Foundation (SF349247) and NY State Assembly. Cryo-EM data processing was performed utilizing the computing resources available at NYU School of Medicine's High-Performance Computing (HPC) facilities.

## Author contributions

J.L. performed NorA$^{E222Q,D307N}$ nanodisc reconstitution for Fab generation, screened Fab hits from phage display for binding to NorA, prepared samples for cryo-EM, processed and analyzed cryo-EM datasets of NorA, NorA$^{D307N}$ and NorA$^{E222Q}$, built atomic models, interpreted the structures, performed binding assays, designed and performed MIC, growth inhibition experiments and efflux assay in MRSA, carried out immunoblotting analyses, carried out and analyzed the MD simulations, designed the project, and wrote the manuscript. Y.L. froze all grids for cryo-EM, processed and analyzed the cryo-EM dataset of NorA$^{E222Q,D307N}$, built the atomic model of NorA$^{E222Q,D307N}$. A.K. and S.K. screened and identified Fab clones to NorA. H.K. collected cryo-EM datasets. V.J.T. directed the MRSA measurements. D.N.W. directed and designed the project. N.J.T. directed and designed the project and wrote the manuscript. All authors participated in revising the manuscript.

## Competing interests

A.K., S.K., D.N.W. and N.J.T. are listed as inventors of a patent application filed by New York University on NorA inhibitors (WO2023014970A2). S.K. is a co-founder and holds equity in Aethon Therapeutics and Revalia Bio; has received consulting fees from Aethon Therapeutics and Black Diamond Therapeutics; has received research funding from Aethon Therapeutics, Argenx BVBA, Black Diamond Therapeutics, and Puretech Health, all outside of the current work. V.J.T. is an inventor on patents and patent applications filed by NYU, which are currently under commercial license to Janssen Biotech Inc. Janssen Biotech Inc. has provided research funding and other payments associated with the licensing agreement. The remaining authors declare no competing interests.
