## [Peer Review File · Nature Communications]

Proton-coupled transport mechanism of the efflux pump NorAReviewer #1 (Remarks to the Author):

Comments: NCOMMS-23-54668

The manuscript authored by Li et al is in continuation of the earlier research on NorA by the Wang and Traaseth laboratories and they try to gain a mechanistic understanding of NorA by studying the alternate conformational states through protonation and deprotonation events at critical acidic residues in the NorA vestibule. The strategy employed by the authors to obtain the inward occluded structures using protonation mimicking substitutions and a new Fab specific for the inward open state is commendable and they come up with an elegant mechanism of how protonation sites aid in the transport mechanism of NorA.

The reported resolutions are generally enough to visualise side chains in the transporters although it would have been better if the authors had provided coordinates and maps for me to visualise the structures particularly to check the densities at important sites. I hope this can be arranged in the revision of the manuscript.

I have the following comments with regards to the manuscript which require to be addressed by the authors.

1. I have an issue with the authors claiming to provide a general mechanism for antiport. While the mechanism they observe maybe true for NorA it is not clear if this holds true for other well-known DHA transporters like mdfA, lmrP and DHA2 members like QacA. One basic issue is the different numbers of protonation sites that are present in each of these unique transporters and different types of substrates that they are capable of transporting. The authors should be a little more measured in their discussion and suggest that this behavior is consistent with fluoroquinolone efflux in NorA rather than associate it with a wider number of antiporters. If they do intend to make this a wider mechanistic principle they have to experimentally demonstrate this in MdfA and LmrP to say the least which would be challenging to accommodate in this manuscript.
2. The description of motifs that are unique to MFS transporters and antiporters are missing in the manuscript. Particularly it would be nice to know that shifts in the motif B and motif C during the reorganization of the transporter from outward open to inward-occluded conformations in NorA.
3. Page 6, Observed rotameric shift of F140- define the extent of shift in torsion angles in both text and figure. The information is missing in the current version.
4. The authors use ethidium bromide a monovalent cation to perform their assays for electrogenic transport. Since this transporter is a fluoroquinolone antiporter it would be nice to show the changes in affinity of NorA with an FQ in the wild-type, single and double mutant constructs.
5. A reference to "belt and suspenders" is likely something very specific to NorA protonation sites and unlikely to be a common theme for antiporters. A schematic in figure 4 can be introduced to improve the clarity of their mechanism. Discussion maybe suitably modified.
6. Extended data figure 1- In these plots, the symbols are incorrect. In legend, they have made grey circles however in the plots, it is grey triangles.
7. The NTD seems to undergo a greater level of rocker movement leading to asymmetry. The authors may comment on the different extents of this movement and this was also recently suggested with the QacA structure.
8. Figure 1. Surface cutaways may be shown with the electrostatics in the vestibule to demonstrate the inward-occluded state of the mutant structures.
9. Figure 3. Pls show the gating residues that weaken in the inward-occluded state and yield inward open NorA during the simulation and allow solvent access. Cytosolic rim mutants are known to play an

important role in (<https://doi.org/10.1016/j.jmb.2018.02.026>) dictating inward-open conformation in MdfA and the figure and text in this portion do not reflect any of these details.

10. The theoretical pKas of the two residues E222 and D307 are not reported by the authors. This would be important to know the sequence of deprotonation events of residues and whether both are simultaneously deprotonated.

11. Also the functional assays reported by the authors are not enough to substantiate the mechanism. The reason for this is because E222Q seems to display some activity over delta-NorA in figure 5b. Rather than growth assays, the authors should try a direct efflux assay typically done for mdfA and QacA by monitoring ethidium efflux as a measure of efflux activity. This is important as the roles of individual protonation sites might be different as seen in other members as one site might be used for protonation to enhance H⁺:substrate stoichiometry and the other site may be a more direct substrate recognition site. Growth assays can be used as supportive data to direct efflux measurements.

Reviewer #2 (Remarks to the Author):

The authors used cryo-EM to elucidate the structures of NorA, NorA (E222Q,D307N), NorA (E222Q), and NorA (D307N) in this work, together with molecular dynamics simulations and binding assays, they proposed a proton-coupling mechanism of the *Staphylococcus aureus* efflux pump NorA. It is a critical piece to complete the puzzle of NorA. Here are some comments and concerns for this manuscript.

Major concerns:

1. In the "MRSA growth inhibition assays support the tethering model" section, the authors have presented several hypotheses without providing sufficient experimental evidence to substantiate their claims. To enhance the study's credibility, it is imperative to incorporate robust experimental data that unequivocally supports each stated hypothesis. By presenting concrete evidence, the authors can strengthen the validity of their assertions and ensure a more reliable foundation for the proposed tethering model.
2. The assertion that "Protonation of Glu222 or Asp307 independently stabilizes the occluded state" appears contradictory to the findings presented in Figure 5b, where D307N almost malfunctions NorA. Notably, numerous residues within a 5Å proximity could influence the function of NorA, therefore, a more comprehensive understanding of the transport process is warranted, considering the multifaceted interactions revealed by the experimental data.
3. While acknowledging the inherent challenges, obtaining structures of NorA in complex with proton, drug substrate, or in the outward-occluded state would significantly bolster the persuasiveness of our findings. These structural insights would provide a more comprehensive understanding of the molecular interactions involved and enhance the overall robustness of our conclusions.
4. In Figure 4c and the corresponding section in the main text, it is mentioned that 'Since ΔpH is effectively zero at an external pH value of ~ 7.8 .' It is important to note that diverse cell types employ distinct and sophisticated mechanisms to regulate intracellular pH. Therefore, the generalization of the ~ 7.8 rule may not be universally applicable to your specific case. Considering the variability in cellular pH homeostasis mechanisms, relying solely on external pH may not be optimal for your study. Moreover, acidic conditions need to be included in this study as well.

Minor concerns:

1. Why some side chain density in the NorAD307N cryo-EM map be missed?
2. Why use ethidium bromide instead of norfloxacin in some experiments? Likewise for norfloxacin.
3. In Figure 4, the proposed model, what is the signal for deprotonation of Glu222 and Asp307?
4. Figure 4d is very confusing to me, the cycled number and the hybrid draws.
5. In Figure 5b, I suggest using an equal number of samples for each mutant. Figure 5c shows that I141A is less than other mutants.

Reviewer #3 (Remarks to the Author):

This manuscript initially seeks to examine the impact of pH on the structural conformation of the multidrug efflux pump NorA using a combination of cryo-EM, mutagenesis and MD simulations. It is well established that antimicrobial efflux by this family of transporters is proton-dependent. They show that the conformation at pH 5 differs to that at pH 7.5. they identify two key acidic residues, E222 and D307, which when conservatively mutated to E222Q and D307N, produce a conformational change that gives an inward-open state. They collect cryo-EM data on the WT NorA at pH 5, as well as the double mutant, the E222Q single mutant, and D307N single mutant, all at pH 7.5 and show the conformation of the double mutant is consistent with that of the WT NorA at pH 5, providing strong support for the role of these two residues in the conformational cycling of NorA.

This is supported by MD simulations.

Overall, this work is significant and noteworthy.

The methodology is sound. The work is carried out to a high standard, the scientific logic is easy to follow and the results are well justified. There are no notable flaws in the interpretation or conclusions drawn. The simulations appear to have been rigorously carried out and the data analysis is appropriate. The cryo-EM data analysis appears appropriate, although this is outside my area of expertise.

There is sufficient detail for the work to be reproducible. My one criticism would be the computational methods. The text should be revised so that it is easier to follow and to clearly identify which simulation package was used, how long each production simulation was, and how many replicates were performed. At the moment, the logical structure to this section needs work. This information is in the text, but it is currently buried in the details and not easy to follow. Also, it appears that the authors only simulated the WT NorA, with E222 and D307 deprotonated. Did they consider incorporating the conservative mutations, E222Q and D307N, into the simulation as a comparison? While the current simulations are sufficient to support the cryo-EM and conclusions, it would have been good to see some additional data on this.

We thank the reviewers for their positive comments and suggestions for improving the manuscript. Below please find our point-by-point responses displayed in bold text.

Reviewer #1 comments

The reported resolutions are generally enough to visualise side chains in the transporters although it would have been better if the authors had provided coordinates and maps for me to visualise the structures particularly to check the densities at important sites. I hope this can be arranged in the revision of the manuscript.

We included a link to download cryo-EM maps and structures in the cover letter to the editor. The editor indicated this is the preferred method for sharing the data for review.

1. I have an issue with the authors claiming to provide a general mechanism for antiport. While the mechanism they observe maybe true for NorA it is not clear if this holds true for other well-known DHA transporters like mdfA, lmrP and DHA2 members like QacA. One basic issue is the different numbers of protonation sites that are present in each of these unique transporters and different types of substrates that they are capable of transporting. The authors should be a little more measured in their discussion and suggest that this behavior is consistent with fluoroquinolone efflux in NorA rather than associate it with a wider number of antiporters. If they do intend to make this a wider mechanistic principle they have to experimentally demonstrate this in MdfA and LmrP to say the least which would be challenging to accommodate in this manuscript.

We agree with the reviewer and revised the text to remove the words “general” and “universal”. While we hypothesize this mechanism may be shared in other transporters, we do not have evidence in other systems and therefore focused our discussion on NorA rather than extrapolating the model to other antiporters.

2. The description of motifs that are unique to MFS transporters and antiporters are missing in the manuscript. Particularly it would be nice to know that shifts in the motif B and motif C during the reorganization of the transporter from outward open to inward-occluded conformations in NorA.

We included structural views of motifs A, B, and C of NorA which compare the inward-occluded and outward-open conformations (Extended Data Figure 5b). As written in the text, we observed that “the interaction between Asp63 in TM2 (part of motif A) and Arg324 in the linker between TM10 and TM11 was disrupted in the inward-occluded state relative to that in the outward-open state (Extended Data Fig. 5b). Furthermore, Arg98 in TM4 (part of motif B) displayed interactions within 4 Å to backbone carbonyls of Gly18 and Ile19 in TM1 for the inward-occluded state that were disrupted in the outward-open conformation. We also observed a more pronounced bending of TM5 at Gly143 and Pro144 of motif C for the inward-occluded state compared to the outward-open structure” (Page 4).

3. Page 6, Observed rotameric shift of F140- define the extent of shift in torsion angles in both text and figure. The information is missing in the current version.

In the updated manuscript, we report the change in the chi1 torsion angle (~67°) for F140 in the text and in Extended Data Figure 5f.

4. The authors use ethidium bromide a monovalent cation to perform their assays for electrogenic transport. Since this transporter is a fluoroquinolone antiporter it would be nice to show the changes in affinity of NorA with an FQ in the wild-type, single and double mutant constructs.

Although we tried to develop a direct binding assay between NorA and fluoroquinolones, such as norfloxacin, we were unable to obtain reliable binding data. We think there are two reasons for this. First, norfloxacin is poorly soluble in aqueous solutions, which is exacerbated by likely poor binding affinity to NorA. Second, norfloxacin's hydrophobic nature leads to nonspecific interactions with detergents or lipids, complicating the accuracy of binding affinity measurements. The

phenomenon of nonspecific interactions affecting binding affinity is a common obstacle in the study of drug transporters, especially when dealing with hydrophobic drugs. An example of this is the strong interaction between the hydrophobic drug tetracycline and detergents or lipids, as noted in the literature (Yung-Chih et al, *Journal of Chromatography A* 1998, 802(1):95-105; Jennie et al, *ACS Omega* 2023, 8(32): 29314-29323; Phillip et al, *Anal Chem* 2011, 83(23):8877-85; Lawrence et al, *Journal of Pharmacokinetics and Biopharmaceutics* 1980, Vol. 8, No. 6; Filipa et al, *Journal of Pharmaceutical Sciences* 2013, 102:1504–1512; Mecheri et al, *Biophysical Chemistry* 2004, 111 (1): 15-26).

5. A reference to “belt and suspenders” is likely something very specific to NorA protonation sites and unlikely to be a common theme for antiporters. A schematic in figure 4 can be introduced to improve the clarity of their mechanism. Discussion maybe suitably modified.

We composed a schematic of the “belt and suspenders” mechanism in a new figure (Figure 4). With regard to describing this model, we focused our discussion on NorA and did not extrapolate the model as universal for explaining other antiporters.

6. Extended data figure 1- In these plots, the symbols are incorrect. In legend, they have made grey circles however in the plots, it is grey triangles.

We corrected the legend in Extended Data Fig. 1 in the resubmitted manuscript.

7. The NTD seems to undergo a greater level of rocker movement leading to asymmetry. The authors may comment on the different extents of this movement and this was also recently suggested with the QacA structure.

Superimposition of TM1-6 between the inward-occluded and outward-open conformations of NorA displays an r.m.s.d. of 1.268 Å, whereas the same overlay for TM7-12 displays an r.m.s.d of 1.458 Å. Given this observation and the fact that our MD simulations did not probe the transition between inward- and outward-facing conformations, we are unable to conclude whether the N-terminal domain movement is more responsible for the rocker-switch mechanism than the C-terminal domain. Regardless, the ~50° change between the N- and C-terminal reported in the manuscript supports the conclusions of a rocker-switch mechanism.

8. Figure 1. Surface cutaways may be shown with the electrostatics in the vestibule to demonstrate the inward-occluded state of the mutant structures.

We revised Figure 1 to display the cutaway views of the electrostatic surface of the inward-occluded state. These views indicate that D307 is more solvent accessible while E222 is more buried in the substrate binding pocket. Additionally, surface representations of the NorA mutants in the inward-occluded conformations are shown in Figure 4.

9. Figure 3. Pls show the gating residues that weaken in the inward-occluded state and yield inward open NorA during the simulation and allow solvent access. Cytosolic rim mutants are known to play an important role in (<https://doi.org/10.1016/j.jmb.2018.02.026>) dictating inward-open conformation in MdfA and the figure and text in this portion do not reflect any of these details.

We included a description of gating residues in the text and a structural view in Extended Data Figure 8h. We also cited the MdfA paper in the manuscript. Namely, we wrote the following: “In addition to changes at Glu222 and Asp307, we observed residues Lys125, Phe129, Ser318 and Asn325 on the cytoplasmic face (termed the “cytoplasmic rim”) display altered interactions upon pocket opening that suggested a role in gating between inward-occluded and inward-open conformations” (Page 8).

10. The theoretical pKas of the two residues E222 and D307 are not reported by the authors. This would be important to know the sequence of deprotonation events of residues and whether both are simultaneously deprotonated.

We agree that the sequence of deprotonation events is an interesting question. According to our inward-occluded structures, D307 is more solvent accessible while E222 is buried within the

substrate binding pocket (see Figure 1). Using PROPKA 3.0, we calculated that D307 has a more acidic pKa value than E222 by ~3 pH units. Although this value correlates with the relative solvent accessibility suggesting D307 deprotonates first, our preference is to measure pKa values experimentally for several conformations of NorA and leave this discussion to future work. As such, we did not introduce this into the manuscript.

11. Also the functional assays reported by the authors are not enough to substantiate the mechanism. The reason for this is because E222Q seems to display some activity over delta-NorA in figure 5b. Rather than growth assays, the authors should try a direct efflux assay typically done for mdfA and QacA by monitoring ethidium efflux as a measure of efflux activity. This is important as the roles of individual protonation sites might be different as seen in other members as one site might be used for protonation to enhance H⁺:substrate stoichiometry and the other site may be a more direct substrate recognition site. Growth assays can be used as supportive data to direct efflux measurements.

We performed ethidium efflux assays in MRSA^{ΔnorA} transformed with empty vector (ΔnorA) or a plasmid harboring wild-type NorA or mutants at E222 and D307 (Figure 6d, e). Efflux data for E222Q and D307N mutants displayed a significant loss of efflux activity compared to wild-type NorA. These results are consistent with our MIC results in Figure 6b and provide additional support for our proposed tethering mechanism.

Reviewer #2 comments

1. In the "MRSA growth inhibition assays support the tethering model" section, the authors have presented several hypotheses without providing sufficient experimental evidence to substantiate their claims. To enhance the study's credibility, it is imperative to incorporate robust experimental data that unequivocally supports each stated hypothesis. By presenting concrete evidence, the authors can strengthen the validity of their assertions and ensure a more reliable foundation for the proposed tethering model.

In this section, we performed functional experiments to test our tethering model hypothesis. We first discuss MIC measurements for E222Q and D307N mutants of NorA. Namely, these mutants displayed nearly total loss-of-function relative to wild-type NorA, which supported binding experiments of wild-type NorA and E222Q and D307N mutants to FabDA1 and Fab36 and the cryo-EM structures showing mutants in the inward-occluded conformation. In this revision, we performed new experiments to measure ethidium efflux in MRSA with the goal of further solidifying our conclusions. MRSA^{ΔnorA} was transformed with empty vector (ΔnorA) or a plasmid containing wild-type NorA or with the E222Q or D307N mutants. We observed a significant reduction in efflux activity for E222Q and D307N relative to wild-type NorA.

Following our discussion of functional data for E222Q and D307N, we discuss MIC results for other mutants located close to E222 and D307. We found that N137A displays a notable loss-of-function, which correlates to cryo-EM structures that show close contacts between the side chains of N137 and protonated D307 for the inward-occluded structure. Mutation to alanine would disrupt these electrostatic interactions, thereby destabilizing the inward-occluded conformation and altering the transport cycle. Likewise, we discuss MIC results for the I141Q and I141A mutants. We observed a rather striking phenotype difference between these two mutants. Preservation of the hydrophobic character of the native I141 residue (mutation to alanine) displayed only a modest reduction in activity, while mutation to a polar residue (glutamine) led to a significant loss-of-function. These data are consistent with MD simulation results showing that deprotonation of E222 leads to pocket opening through introduction of water into the substrate binding pocket. Namely, if I141 was changed to a polar residue like the Q141 mutant, the mutant would likely be able to form hydrogen bonds with the ionized E222 and alter the extent of pocket opening. This has been clarified in the resubmitted manuscript within section "*MRSA growth inhibition and efflux assays support the tethering model*".

Overall, the MIC growth and efflux assay results support our conclusions in the manuscript about the tethering model mechanism.

2. The assertion that "Protonation of Glu222 or Asp307 independently stabilizes the occluded state" appears contradictory to the findings presented in Figure 5b, where D307N almost malfunctions NorA. Notably, numerous residues within a 5Å proximity could influence the function of NorA, therefore, a more comprehensive understanding of the transport process is warranted, considering the multifaceted interactions revealed by the experimental data.

Our binding and structural data indicate that E222Q and D307N mutations trap NorA in the inward-occluded conformation. This prevents the transporter from proceeding through all the conformations required of the catalytic cycle (Figure 5d). In other words, if a transporter could only sample the inward-occluded conformation, it would be stuck and could not transport substrates across the membrane. Therefore, the MIC results for the D307N mutant reinforce the proposed transport mechanism. To better convey this finding, in the revised paper, we changed the title of the corresponding Results section from "*Protonation of Glu222 or Asp307 independently stabilizes the occluded state*" to "*Protonation of Glu222 or Asp307 independently traps the occluded state.*" (Page 5)

3. While acknowledging the inherent challenges, obtaining structures of NorA in complex with proton, drug substrate, or in the outward-occluded state would significantly bolster the persuasiveness of our findings. These structural insights would provide a more comprehensive understanding of the molecular interactions involved and enhance the overall robustness of our conclusions.

The current manuscript is focused on elucidating the role of proton-binding and resulted in the elucidation of four cryo-EM structures of NorA in various proton-bound states of E222 and D307. The structures, binding assays, and functional data, including new ethidium efflux assays, support the conclusion that protonation of E222 and D307 favors the inward-occluded conformation of NorA. While determining structures of NorA in other conformations is a long-term goal of this project, this is a significant effort that cannot be easily carried out within a reasonable time frame.

4. In Figure 4c and the corresponding section in the main text, it is mentioned that 'Since ΔpH is effectively zero at an external pH value of ~ 7.8 .' It is important to note that diverse cell types employ distinct and sophisticated mechanisms to regulate intracellular pH. Therefore, the generalization of the ~ 7.8 rule may not be universally applicable to your specific case. Considering the variability in cellular pH homeostasis mechanisms, relying solely on external pH may not be optimal for your study. Moreover, acidic conditions need to be included in this study as well.

To address the reviewer's point, we performed a growth inhibition experiment at pH 6.5 and included this result in the new version (see Figure 5c and Extended Data Figure 9).

The reviewer is correct that intracellular pH is not a fixed value, and different cells regulate it with complex mechanisms. Nevertheless, in our study, we conducted growth assays in the neutrophilic bacterium *S. aureus*, which is reported to have a cytoplasmic pH ranging from 7.4 to 7.7 (Manisha et al., *BioRxiv* 2019; Etana et al, *Biochimica et Biophysica Acta (BBA) – Biomembranes* 1981, 650(2-3): 151-166; Etana et al, *Biochimica et Biophysica Acta (BBA) – Biomembranes* 2005, 1717(2): 67-88; Slonczewski et al., *Advances in Microbial Physiology* 2009, 317:1-79; Ian et al, *Microbiology and Molecular Biology Reviews* 1985, 49(4): 359-378). When assays are performed in growth media at pH 7.0, ΔpH is slightly positive (ΔpH is 0.4 to 0.7). However, when the pH is raised to basic values (i.e., in our case ranged from 6.5 to 8.5), ΔpH transitions from a positive to a negative value. This shift indicates that it is unable to serve as an energy source for ethidium efflux. Consequently, NorA cannot perform ethidium efflux if it employs 1:1 H⁺:ethidium electroneutral transport under these conditions. To make this point more clear, we revised the sentence "*Since ΔpH is effectively zero at an external pH value of ~ 7.8 , the resistance phenotype conferred by NorA at this pH value indicated the membrane potential alone is sufficient to drive transport*" to "*Since ΔpH is zero or negative when the external pH is at or above 7.8, the resistance phenotype conferred by NorA under these conditions indicated the membrane potential alone was sufficient to drive transport*" (Page 9).

Minor concerns:

1. Why some side chain density in the NorAD307N cryo-EM map be missed?

In the cryo-EM analysis of the NorA^{D307N}-FabDA1 complex, we observed more unbound FabDA1 and therefore less intact NorA^{D307N}-FabDA1 complexes. Consequently, we obtained fewer total particles (136,284 particles) for reconstruction of the cryo-EM map compared to other complexes. It is likely this contributed to missing side chain density. Additional factors such as greater intrinsic heterogeneity of NorA^{D307N} may also play a role in the missing density. Regardless, the missing density was not for residues in the substrate binding pocket and therefore none of the conclusions are influenced.

2. Why use ethidium bromide instead of norfloxacin in some experiments? Likewise for norfloxacin.

Ethidium bromide was selected for the growth assay at various external pH values since it has a stable +1 charge over the pH range. Norfloxacin is predominantly neutral at pH 7.0 with two pKa values near the pH range used in our experiments (pKa1 = 6.22, pKa2 = 8.51. Weiben et al, *Chemical Engineering Journal* 2012 179: 112–118). This means it could not be used to distinguish electrogenic and electroneutral transport.

3. In Figure 4, the proposed model, what is the signal for deprotonation of Glu222 and Asp307?

In the updated version (see Figure 5d), the notation for the deprotonated forms of Glu222 and Asp307 has been revised to E222⁻ and D307⁻, respectively.

4. Figure 4d is very confusing to me, the cycled number and the hybrid draws.

We replaced the surface representations in the model with cartoon representations (see Figure 5d). Numbering of steps in the transport cycle was retained to facilitate a simpler description in the figure legend.

5. In Figure 5b, I suggest using an equal number of samples for each mutant. Figure 5c shows that I141A is less than other mutants.

MIC measurements of MRSA^{ΔnorA} transformed with the NorA variants were each independently repeated twice. The MRSA^{ΔnorA} negative control was subjected to a greater number of repeats (5 total) as it was included in every set of experiments, including optimization of the assay. The number of repeats conducted for each mutant is indicated on the top of each corresponding bar by the number of black dots, as required by Nature publishing group when the data points in bar graphs is less than or equal to 10.

The western-blot data indicates a slightly lower expression level for the I141A mutant; however, we do not feel this changes our interpretation of these MIC data in terms of the mechanistic conclusion. Namely, I141Q displays a greater expression level than I141A, yet I141Q shows a significantly lowered MIC value (i.e., loss-of-function). If the lower expression of I141A had an impact on the MIC values, if anything it would have displayed a lower MIC value than I141Q. The hypothesis we propose in the manuscript is that such a hydrophobic residue may be needed at position 141 to encourage opening of the substrate binding pocket. We predict that the I141Q mutation may trap the inward-occluded conformation and slow down the transport cycle leading to the lowered MIC value.

Reviewer #3 comments

My one criticism would be the computational methods. The text should be revised so that it is easier to follow and to clearly identify which simulation package was used, how long each production simulation was, and how many replicates were performed. At the moment, the logical structure to this section needs work. This information is in the text, but it is currently buried in the details and not easy to follow. Also, it appears that the authors only simulated the WT NorA, with E222 and D307 deprotonated. Did they consider incorporating the conservative mutations, E222Q and D307N, into the simulation as a comparison? While

the current simulations are sufficient to support the cryo-EM and conclusions, it would have been good to see some additional data on this.

The Methods section describing the molecular dynamics simulations was revised for clarity and comprehensiveness in the revised version of the manuscript, including the length of production runs and replicates performed for each simulation.

It is a good question about why we didn't perform simulations on E222Q and D307N mutations. While we considered it, the primary reason we ultimately decided not to perform these simulations was that we felt the similarity of the cryo-EM structures for wild-type NorA at low pH and the E222Q/D307N double mutant validated the use of glutamine and asparagine as suitable proton mimics. While experiments required the use of mutants for investigating individual protonation of E222 or D307, the value of computation was that protonation states could be explicitly assigned without the need for mutations. Therefore, we preferred to perform the most relevant simulations using wild type NorA to study the singly protonated states rather than rely on the E222Q or D307N mutants. The findings from the MD simulations concerning the single protonation state of NorA are presented in Extended Data Figure 8. We included additional analysis in this figure by displaying residues at the cytoplasmic rim that may play a stabilizing role in the inward-occluded conformation.

Reviewer #1 (Remarks to the Author):

The authors have satisfactorily answered all of my queries. I am satisfied with the manuscript and have no further edits to suggest.

Reviewer #2 (Remarks to the Author):

Most of my concerns have been addressed properly.
No more other questions.